# Development of a Novel Pictorial Questionnaire to Assess Knowledge and Behaviour on Ergonomics and Posture as Well as Musculoskeletal Pain in University Students: Validity and Reliability

**DOI:** 10.3390/healthcare12030324

**Published:** 2024-01-26

**Authors:** Mona Salman, Josette Bettany-Saltikov, Gokulakannan Kandasamy, Garikoitz Aristegui Racero

**Affiliations:** 1Centre for Public Health, School of Health and Life Sciences, Teesside University, Middlesbrough TS1 3BX, UK; 2Centre for Rehabilitation, School of Health and Life Sciences Allied Health Professions, Teesside University, Middlesbrough TS1 3BX, UK; j.b.saltikov@tees.ac.uk (J.B.-S.); g.kandasamy@tees.ac.uk (G.K.); 3Scoliosis and Posture Centre, BSPTS, 08021 Barcelona, Spain; gari@scoliosis.es

**Keywords:** validity, reliability, knowledge, behaviour, ergonomics, posture, musculoskeletal, pain, university, students

## Abstract

Background: Good posture is characterised by neutral spinal alignment with high physiological and biomechanical efficiency together with low stress and strain. The purpose of this study was to assess the validity and reproducibility of the adult version of the Aristegui questionnaire in university students. Materials and methods: The study was conducted in two parts. The first part assessed content validity of the questionnaire where five experts provided their feedback on the content of the questionnaire. The second part evaluated the reliability of the questionnaire among a convenience sample of 10 university students in a test–retest study. Results: The content validity of the questionnaire was found to be excellent. Twenty-five out of twenty-seven items had an item content validity index higher than 0.79 (appropriate) and the scale content validity index was 0.93 (high). For the reliability, almost perfect agreements were found for nine items, substantial agreement for three questions, moderate agreement for one item and fair agreement for one item. The kappa coefficients ranged from 0.00 (slight) to 1.00 (perfect) for the items on behaviour. Conclusions: The questionnaire was found to be a valid and reliable tool to measure the university students’ knowledge regarding ergonomics and posture and postural behaviour as well as prevalence of musculoskeletal pain.

## 1. Introduction

Good posture is characterised by neutral spinal alignment with high physiological and biomechanical efficiency together with low stress and strain [1,2]. Any deviation from normal alignment can result in nerve compression and abnormal stress on ligaments, joints and cartilages, subsequently leading to musculoskeletal discomfort [3]. A good level of health knowledge can improve an individual’s self-care and therefore leads to better physical and mental health [4]. For instance, the awareness of the most adequate body posture may prevent musculoskeletal pain [5]. Attempts at increasing the awareness of university students regarding back health together with adequate postural behaviour need to be given much higher priority on the national agenda [6]. 

Questionnaires are the most commonly used instruments for collecting data on health and in surveys [7,8]. The items included in a measurement instrument need to be related to the objectives of the study and it is essential that they are appropriate to answer the research question [9,10].

A comprehensive review of the literature was performed on 15 databases (the Cumulative Index to Nursing and Allied Health Literature (CINAHL), MEDLINE, Allied and Complementary MEDicine (AMED), SportDiscus, Embase, Cochrane Central Register of Controlled Trials (CENTRAL) in the Cochrane Library, Campbell Collaboration, Physiotherapy Evidence Database (PEDro), the Education Resources Information Centre (ERIC), Web of Science, SCOPUS, PROSPERO, ETHOS, WHOLIS and ProQuest Dissertations and Theses Global) to identify relevant studies published till 2023. The search terms used in the search strategy were related to “back care”, “education”, “knowledge”, “behaviour”, “ergonomics”, “posture”, “musculoskeletal pain” and “university students”. The review revealed that 21 instruments had been developed to measure knowledge regarding ergonomics and posture as well as postural behaviour. Out of the 21 tools found, 15 were designed for university students [11,12,13,14,15,16,17,18,19,20,21,22,23,24,25]. The rest were primarily targeted at schoolchildren [26,27,28,29,30,31].

Elsallamy et al. [11] used a 16-item questionnaire to explore the knowledge and practice of 479 dental students regarding the ergonomics principles used within dental clinics. The results demonstrated that only 48.9% of the participants had fair knowledge regarding ergonomics and 5% of students practice it. In a similar way, Cervera-Espert et al. [12] carried out a cross-sectional study involving 336 dental students to explore students’ knowledge of ergonomics and posture as well as their postural behaviour. A self-administered questionnaire comprising 32 questions was used to collect data on basic knowledge of ergonomics (questions 1–5) as well as postural behaviour (questions 15–31). However, the authors in both studies were not interested in studying the respondents’ level of knowledge and practice regarding posture whilst using a computer together with carrying a bag and lifting weight. We believe that these aspects are vital to university students’ daily life activities. 

Kanaparthy et al. [13] also conducted a study on dental students (n = 134) but assessed solely the students’ postural behaviour. The participants completed a close-ended questionnaire inquiring about their body postures whilst working in dentistry. In contrast, Movahhed et al. [14] used an 18-item questionnaire to measure knowledge regarding the principles of dental ergonomics in 103 dental students. The questionnaire, which focused only on knowledge, consisted of multiple choice and true/false questions. It is important to note that dental ergonomics might not be relevant to other university students such as those studying computer science, art, psychology, business and so on. This reduces the external validity of the findings obtained from studies targeting only dental students. As a result, the generalisability of these results to the university student population is compromised.

Using a different study sample, López and Martínez [15] evaluated the impact of an educational intervention on body awareness and frequency of injuries in music students (n = 146). The respondents completed a self-administered questionnaire inquiring about the warm-up exercises before practicing their instruments, the practice time, breaks during practice as well as the frequency and duration of the breaks. The students were also asked about any physical discomfort resulting from practicing the musical instrument including the description, duration and location of any symptoms as well as the treatments used to reduce these symptoms. It is important to note that not all university students use musical instruments. This therefore means that posture during musical practice may not be relevant for students in the faculties of health and life sciences, computer science and business. 

More recently, Maghbouli et al. [16] conducted a pre–post design study to evaluate the impact of a back care education programme on the postural behaviour of 27 nursing students. The study participants were administered a questionnaire about body posture on a 3-point Likert scale (always, sometimes and never). However, Maghbouli et al. [16] did not report sufficient information regarding the number of tool items as well as the constructs measured. Thus, it was not possible for the researchers to evaluate whether the instrument they used is appropriate to achieve the aim of the current study or not.

On the other hand, Joshi et al. [17] assessed the knowledge of 60 students studying agriculture and technology about ergonomics related to a computer workstation. The 3-point Likert scale (agree, disagree, undecided) questionnaire asked about the ergonomic workstation (tilt tray–keyboard arrangement, workstation backrest, height of table and chair, monitor position according to eye level, distance between the operator and the monitor) and visual display terminal (computer screen and accessories) as well as the working posture (elbow angle, lumbar support). The authors observed that the majority (>50%) of the students did not have knowledge about correct positioning of computer workstations. 

Using the same sample size, Kamaroddin et al. [18] had earlier asked 60 students from the Faculty of Computer Science and Mathematics about their posture whilst working on computers (chair, keyboard, mouse, monitor and desk). The results revealed that all the participants knew the principles of ergonomics but only half of them practiced it. Additionally, Sirajjuddin and Siddik [19] carried out a cross-sectional study to assess the knowledge of computer ergonomics on a large sample (n = 177) of computer science engineering and information technology students. The participants completed a valid and reliable questionnaire comprising 35 items related to knowledge about MSK disorders and their risk factors and sitting postures as well as positioning of the keyboard/mouse, monitor and table. Taken together, the above-mentioned studies focused only on the knowledge of posture and ergonomics needed for working at a workstation, neglecting the importance of the posture the university students adopt when carrying a bag which is an essential part of their academic life. The authors also did not assess students’ postural behaviour which is one of the outcomes of the current study.

Previous studies such as those of Dolen and Elias [20] and Bowman et al. [21] focused on the knowledge and behaviour of laptop ergonomics (posture of the head, neck, back, elbow, wrist and hand) in university students. In the study by Dolen and Elias [20], 197 students (101 health science and 96 economics and management students) completed 24 yes/no items on knowledge and 40 Likert-scale type questions on practice of laptop ergonomics. A total of 74.1% students were found to have a fair knowledge on laptop ergonomics with 15.8% having poor knowledge. Further, 70% of the students had fair practice in laptop ergonomics and 16.8% had good practice. However, the authors did not report any details regarding the items and did not establish the validity of the instrument they used. Thus, it was not possible to assess whether the 64 items were relevant to the current study or not. Additionally, a long questionnaire may reduce the response rate due to the long time needed to complete it.

In a study that investigated the effect of a laptop ergonomic education intervention on university students’ knowledge and behaviours regarding proper laptop use, Bowman et al. [21] included 172 university students (83 occupational therapy, 63 physical therapy and 26 nurse anaesthesia). The completed questionnaire comprised seven multiple choice questions, two true/false questions and seven pictures in which participants identified “good” or “bad” laptop ergonomics. The results demonstrated a statistically significant improvement in ergonomics knowledge of the experimental group after attending the educational programme. Although computers and laptops constitute an important component of ergonomics and body posture, other vital features (carrying backpacks, lifting objects and sitting) were not addressed in the above-mentioned studies.

Jaafar et al. [22] asked 246 engineering students whether or not they were aware of the benefits of ergonomics together with the negative health consequences of not practicing ergonomics principles. Furthermore, Jacquier-Bret and Gorce [23] conducted a cross sectional study to assess the effect of the time of day on the posture adopted during smartphone use among 263 university students. Similarly, a study by Odole et al. [24] investigated patterns of musculoskeletal pain, postural abnormalities and smartphone usage among 400 undergraduate university students. Gorce and Jacquier-Bret [25] examined how the duration of smartphone use varies by the time of day and activities and the risks of MSDs based on an analysis of the postures used when interacting with smartphones in a sample of 263 university students. However, all three studies were fully focused on postures adopted while using smartphones which were not considered to be part of our study.

Therefore, it can be concluded that the constructs and case scenarios used in questionnaires targeting university students are narrow in focus and did not suit the purpose of the current study. The knowledge regarding ergonomics and posture and postural behaviour (whilst sleeping, lifting objects, carrying backpacks, taking objects from a high shelf, sitting at a desk, on an armchair and at a table and using computers) were not fully covered in all the above-mentioned tools.

Furthermore, the validity of previous data collection tools was not established in eight studies [12,13,15,16,17,20,21,22]. Hence, the above-mentioned tools cannot be considered valid instruments to measure knowledge of ergonomics and posture as well as postural behaviour in university students. 

In comparison with the instruments described above, the child version of the Artisegui questionnaire [29] consisted of three multiple choice questions on the presence of back pain, nine questions on knowledge as well as nine questions on behaviour with illustrated pictures of the students’ posture: lying down, lifting a weight, taking objects from a high shelf, sitting (desk, armchair, table) and using a computer as well as carrying a backpack and bag on wheels. The child version of the Artisegui questionnaire was developed to assess the effect of an educational intervention on postural habits and the prevention of back pain amongst Spanish schoolchildren aged 8–10 years old. The questionnaire was based on a previous questionnaire [32] where ten items were extracted and translated into pictures by Artisegui [29] together with an artist to ensure better understanding of the items by schoolchildren aged 8–10 years old. Unfortunately, Aristegui [29] did not assess the psychometric properties of the final version of the questionnaire.

To the researchers’ knowledge, none of the available questionnaires listed above can be considered to be valid instruments for measuring knowledge regarding ergonomics, posture and postural behaviour in university students. However, the child version of the Aristegui questionnaire was specifically related to knowledge regarding ergonomics and posture as well as postural behaviour, illustrated with images, which was highly original. Thus, the Aristegui tool (child version) was considered to be the most appropriate instrument to evaluate the knowledge regarding ergonomics and posture as well as postural behaviour in university students. 

The lack of validation of this questionnaire in schoolchildren as well as in the university population highlighted the necessity of establishing its validity. It is essential for researchers to assess the reliability of a new data collection tool before using it in a research study [33]. Reliability is defined as the stability of responses if the same instrument is used at times [34]. A reliable research instrument provides more precise results upon which correct decisions can be made [35]. Additionally, the data collection tool is more precise when the results of measurements are closer [36]. 

Since most of the above-mentioned studies have not tested the validity and reliability of the instruments they used, these questionnaires cannot be considered appropriate to measure the knowledge regarding ergonomics and posture as well as postural behaviour in university students. To the best of the authors’ knowledge, there are no reliable instruments designed to assess and gain a deeper understanding of knowledge regarding ergonomics, posture and postural behaviour in university students. Thus, the purpose of this study was to: (1) validate the adult version of the Artisegui questionnaire and (2) assess its reliability using a test–retest study design. Such instruments are important tools that can be used in epidemiological and experimental studies. This can significantly help researchers and healthcare professionals to evaluate the level of knowledge regarding ergonomics, posture and postural behaviour. Therefore, early public health preventive interventions can be created.

## 2. Materials and Methods

In order to analyse the psychometric properties of the instrument, two steps were carried out: (1) content validity and (2) test–retest reliability.

### 2.1. Sampling

Purposive sampling was used to recruit participants for the content validation process. Participants needed to be experts having at least a master’s degree in physiotherapy, sport therapy, nursing, chiropractor or podiatry from the School of Health and Life Sciences at a university in the northeast of England. These subjects were chosen as they include knowledge and experience in the field of ergonomics and body posture. Experts who were involved in the research project (research supervisory team) were excluded from the study to minimise any risk of bias. 

No consensus exists in the literature on how to determine the number of experts for conducting the content validity of a research study. However, the number of experts required to have a good level of chance of agreement is between 5 and 10 judges [37]. Based upon the accessibility of experts and the recommendations for the sample size, 5 experts in physiotherapy, 2 experts in sports therapy, 1 in nursing, 1 chiropractor and 1 podiatrist were asked to judge the content validity of the adult version of the Aristegui questionnaire. 

The test–retest study was conducted on a convenience sample of university students, at a university in the northeast of England, who completed the Aristegui questionnaire (adult version). Participants were eligible for the study if they were students at a university in the northeast of England, including both genders, full-time and part-time attendance, flexible and distance learners and undergraduate and postgraduate students. According to the table in [38], by assuming alpha of 0.05, power of 80%, a medium effect size (K1 0.3 and K2 0.8) and number of responses for the majority of items of 4, the minimum sample size required for the current reliability study was 12. 

### 2.2. Instrumentation

Following an extensive literature review, the child version of the Aristegui questionnaire was considered to be the most appropriate instrument to measure knowledge of ergonomics, posture and postural behaviour in young adults. This 21-item questionnaire was designed with attractive illustrations in an attempt to make the items easier to understand and increase the response rate. Originally, Aristegui [29] designed the tool for schoolchildren aged 8–10 years old. Following the relevant permissions from Aristegui [29] to use these questions, the research team (researcher and 2 supervisors) slightly amended the questionnaire to make it more suitable for university students (adult version) (see Figure 1). 

The research team added 5 items on demographics, an item asking the respondents whether or not they carried a backpack and 9 open-ended questions following the 9 close-ended questions measuring the knowledge of ergonomics and posture. Hence, the adult version of the Aristegui questionnaire comprised 36 items. The research team also replaced “back pain” by “musculoskeletal pain” to provide a wider picture as neck pain and shoulder pain are also considered to be relevant and related to postural health problems. 

The research team also agreed to include open-ended questions to assess whether or not the participants have an in-depth understanding or they had just guessed the answer. The researchers added the options “other pain” to the question on the presence of musculoskeletal pain, “all of the above” and “none of the above” to the questions about postural behaviour and “I don’t know” to the close-ended questions inquiring about knowledge of ergonomics and posture. 

The 36-item instrument consisted of three sections. These comprised: Section 1—Background information (age, gender, course of study, academic year and course level); Section 2—Musculoskeletal pain; Section 3—Knowledge and behaviour of ergonomics and posture. Three items were related to musculoskeletal pain and asked about the presence of pain, the location of pain and the activity after which the participant experienced pain. Nine items were related to postural behaviour and 18 questions (9 close-ended and 9 open-ended) assessed the knowledge of ergonomics and posture. The adult version of the Aristegui questionnaire was sent to experts for validation (27 items) without including the 9 open-ended questions. This is because validity is a term commonly used for quantitative data and not qualitative data.

The instrument version used in the test–retest reliability (see Figure 2) comprised 25 items and consisted of three sections on background information (age, gender, course of study), musculoskeletal pain as well as knowledge and behaviour of ergonomics and posture. Three items were related to musculoskeletal pain and asked about the presence of pain, the location of pain and the activity after which the participant experienced pain. Nine items were related to postural behaviour, one item asked whether or not the participant carried a backpack when attending university and nine questions assessed the knowledge of ergonomics and posture. 

### 2.3. Procedure

The researchers sent the content validation form (see Figure 3) as well as the adult version of the Aristegui questionnaire (see Figure 1) to the experts via email. The content validation form included instructions for the review panel of experts on how to provide their quantitative and qualitative feedback. The content experts were asked to independently score the relevance of each item on the instrument using a 4-point Likert scale (1 = not relevant, 2 = somewhat relevant, 3 = relevant, 4 = very relevant). The relevant Ethics Committee provided approval for this study. The experts were not asked to sign a written consent form as completing the validation form was taken as consent. 

Thirteen students participated in Test 1 of the reliability study. Of these respondents, 10 completed the questionnaire in Test 2. After signing the consent form, the respondents completed the emailed questionnaire. Unique numbers for the study were assigned to each participant so that the researchers were able to match the questionnaires from the test and retest. It took approximately 10 min to complete the questionnaire. After a one-week period, the questionnaire was administered again to the university students who were participating, using the same procedures for Test 1. For the second session, the items were presented in a different order to the same participants to minimise the risk of bias resulting from a memory effect. The present study was approved by the Ethics Committee at a university in the northeast of England. No monetary rewards were given to each respondent at the end of the tests. 

### 2.4. Data Analysis

Data entry and analysis of the content validation study and the test–retest reliability study were performed using SPSS version 26. The content validity of the adult version of the Aristegui questionnaire was evaluated by calculating the I-CVI and the S-CVI as well as the Kendall coefficient of concordance. I-CVI was obtained by dividing the number of experts giving a score of 3 (relevant) or 4 (very relevant) by the total number of experts according to the formula [37]: I-CVI = number of experts rating the item 3 or 4/total number of experts

Items with an I-CVI higher than 0.79 were considered to be appropriate [39]. Items with I-CVI between 0.70 and 0.79 were considered to need revision. Items with I-CVI less than 0.70 were deleted. On the other hand, the S-CVI/Ave is defined as the average of the I-CVI scores for all items on the scale and calculated as follows [37]: S-CVI/Ave = sum of all I-CVIs/total number of items 

The S-CVI/Ave needs to be 0.90 or above in order to be acceptable [40]. As the CVI does not adjust for chance agreement, the Kendall coefficient of concordance and corresponding *p*-value were also calculated. A Kendall W of 0.40 or less represents poor agreement, values between 0.40 and 0.75 indicate fair to good agreement and values of 0.75 or above represent excellent agreement [41]. A corresponding *p*-value of less than 0.05 indicates a statistically significant agreement between the experts. 

For the test–retest reliability, descriptive statistics were used to present the baseline demographic characteristics of participants (age, gender, type of course) and musculoskeletal pain (presence, location and activity after which they experienced pain) as well as the response rate. 

Cohen’s kappa statistic for nominal variables and weighted kappa coefficient with 95% confidence interval for the ordinal variable (age) were computed to estimate the level of agreement for the 10 participants at two time points. The kappa coefficients can be interpreted as follows (41): <0 (poor agreement); 0–0.2 (slight agreement); 0.21–0.40 (fair agreement); 0.41–0.60 (moderate agreement); 0.61–0.80 (substantial agreement); 0.81–1.0 (almost perfect agreement). The 95% confidence interval (CI) and the standard error of kappa (SEκ) for each obtained kappa value were also estimated. The researchers followed the formula below to calculate the 95% confidence intervals [42,43]:CI: k − 1.96 × SEκ to k + 1.96 × SEκ 
where k is the kappa coefficient, 1.96 is a constant and SEκ is the standard error of kappa [43]. 

## 3. Results

### 3.1. Content Validation

Five out of the ten invited content experts agreed to participate in the study and returned the completed form to the researchers through email. Two were experts in sport therapy, one in physiotherapy, one in nursing and one in podiatry. The Kendall W score obtained was 0.46 and the corresponding *p*-value was 0.00, therefore indicating a statistically significant fair to good agreement among the experts. The I-CVIs of each item on the adult version of the Aristegui questionnaire, as shown in Table 1, ranged from 0.4 to 1.00. Out of the 27 items, 25 items had an I-CVI higher than 0.79 and were considered to be appropriate. The remaining two items (Qs4 and Qs5) were eliminated because they demonstrated an I-CVI lower than 0.7. Further, the S-CVI of the whole instrument was found to be 0.93 (high). The details of the calculation are explained as follows: Sum of all I-CVIs = 1 + 1 + 1 + 0.4 + 0.4 + 1 + 1 + 0.8 + 1 + 1 + 1 + 1 + 1 + 1 + 1 + 1 + 1 + 0.8 + 0.8 + 1 + 1 + 1 + 1 + 1 + 1 + 1 + 0.8 = 25S-CVI/Ave = sum of all I-CVIs/total number of items = 25/27 = 0.93 

As a result, the number of items on the instrument was reduced from 27 to 25 questions after the validation process. Therefore, the adult version of the Aristegui questionnaire was found to achieve a good level of content validity. Table 1 presents the results of the content validity study. With regard to the qualitative feedback, all the five content experts agreed that the constructs were good and the wording of the items was clear, unambiguous and appropriate for university students. However, two experts suggested minor amendments to some of the items and the researchers incorporated those amendments into the instrument. The option “walking around” has been added to the other choices in the item about activities after which the participant experienced musculoskeletal pain. Further, an annotation was added to clarify the picture related to the carriage of a bag on wheels. 

### 3.2. Test–Retest Reliability

Out of the 13 students who participated in the test–retest study, 10 completed the retest. Thus, the response rate was good (77%). There were more males (60%) than females (30%) in the sample. The respondents were aged 18 to 23 (10%), 24 to 29 (30%), 30 to 34 (10%) and older than 34 years old (50%). About 20% of respondents studied computer science, 10% biomedical sciences, 10% business, 10% education, 10% electric engineering, 10% law, 10% microbiology, 10% psychology and 10% public health. The demographic characteristics of the respondents who completed the test and retest are presented in Table 2. 

There were no missing data in the completion of the questionnaire for the test–retest reliability. The test–retest reliability results for the adult version of the Aristegui questionnaire are displayed in Table 3. The kappa values of the items in the adult version of the Aristegui questionnaire ranged from 0.00 (slight agreement) to 1.00 (perfect agreement). The consistency between the two tests with regard to the demographic characteristics (age, gender and course of study) was established. The weighted kappa value for the age variable was 1.00 (*p* = 0.00). Similarly, the unweighted kappa coefficients for the items on gender as well as the course of study were 1.00 (*p* = 0.00), indicating perfect agreement. The kappa coefficient for the items related to the presence of musculoskeletal pain (Qs 4) was 1.00 (*p* = 0.00), indicating perfect agreement. For multiple choice questions where the participant could choose more than one answer, each option was entered as a variable into SPSS. The kappa values ranged between 0.60 and 1 for musculoskeletal pain location (Qs 5) and 0.55–1 for the activities after which the participant experienced musculoskeletal pain (Qs 6).

Almost perfect agreements were found for the items evaluating knowledge of adequate body posture when sitting at a desk (Qs 12) (k = 1.00, *p* = 0.00), on an armchair (Qs 19) (k = 0.80, *p* = 0.01) and at a table (Qs 21) (k = 1.00, *p* = 0.01) as well as at a computer (Qs 23) (k = 1.00, *p* = 0.01). Similarly, the item asking whether or not the participant used a backpack (Qs 13) achieved a kappa value of 1.00 (*p* = 0.00). Substantial agreement was shown for the questions related to knowledge of adequate posture when picking up objects from the ground (Qs 10) (k = 0.63, *p* = 0.01) and carrying backpacks (Qs 15) (k = 0.78, *p* = 0.00) as well as taking objects from a high shelf (Qs 17) (k = 0.63, *p* = 0.01). The questions on knowledge of adequate sleeping posture (Qs 8) and posture when carrying a bag on wheels (Qs 25) had kappa values of 0.51 (*p* = 0.03) (moderate agreement) and 0.29 (*p* = 0.19) (fair agreement), respectively. The question which achieved fair reliability (Qs 25) was deleted (k < 0.41, *p* > 0.05). 

For the multiple choice questions eliciting information on postural behaviour, the kappa coefficients were as follows: sleeping (Qs 7) (k = 0.44–1), picking up objects from the floor (Qs 9) (k = 0.00–0.62), sitting at a desk (Qs11) (k = 0.41–0.78), carrying a backpack (Qs 14) (k = 0.62–1), taking objects from a high shelf (Qs 16) k = 0.41–1), sitting on an armchair (Qs 18) (k = 0.00–0.8), sitting at a table (Qs 20) (k = 0.35–0.58), sitting at a computer (Qs 22) (k = 0.52–1) as well as carrying a bag on wheels (Qs 24) (k = 0.00–0.54). The latter question (Qs 24) was also removed as this is directly related to the removed item measuring knowledge of correct posture whilst carrying a bag on wheels (Qs 25). As a result, the number of items of the adult version of the Aristegui questionnaire was reduced from 25 to 23 after the test–retest reliability study.

## 4. Discussion 

The literature lacks valid and reliable tools designed to evaluate knowledge regarding ergonomics and posture as well as postural behaviour. To the best of the authors’ knowledge, this is the first study to assess the validity and reproducibility of the adult version of the Aristegui questionnaire to be used for measuring knowledge regarding ergonomics and posture as well as postural behaviour in university students. The child version of the Aristegui questionnaire was illustrated with images which was highly original. It was developed with the help of an artist who painted the cartoon pictures. Thus, the Aristegui tool (child version) was considered to be the most appropriate instrument to evaluate the knowledge regarding ergonomics and posture as well as postural behaviour in university students. Despite a comprehensive search of the literature, we were unable to find a similar instrument that could improve the engagement of the university students with the questions. 

The results of the validation study demonstrated that the adult version of the Aristegui questionnaire had very good content validity. The overall content validity index of the instrument was found to be 0.93, indicating a high content validity. Among the 27 items on the instrument, only 2 items presented a low I-CVI (<0.7) which were then removed. Based on the experts’ recommendations, minor amendments to a few of the items were performed. Thus, the adult version of the Aristegui questionnaire can be used as a valid tool to measure knowledge regarding ergonomics and posture as well as postural behaviour in university students.

It is important to note that previous studies [26,28] did not use any quantitative methods to assess the content validity of the instruments they used. Thus, the results from the current study could not be compared to their findings. For instance, six experts and ten students evaluated the comprehensibility and the ease of use of the 24-item health questionnaire measuring the adolescents’ level of back care knowledge in activities of daily living [28]. Similarly, six researchers, with expertise in low-back research work and PhDs in physical education, medicine and physiotherapy, together with eight schoolchildren, provided their qualitative feedback on the clarity and relevance of the items in the study by Monfort-Pañego and Miñana-Signes [26]. 

Although both studies by Monfort-Pañego and Miñana-Signes [26] and Monfort-Pañego et al. [28] took into account the opinion of both the target population and the experts, they relied solely on qualitative feedback. Quantitative and qualitative methods, if used together, may provide a more comprehensive evaluation of the content validity of the instrument used. A major strength of the mixed methods approach is that it results in more complete and rich data compared to each method alone [44]. 

On the other hand, the opinion of judges with good knowledge and experience in the field of ergonomics and posture, such as healthcare professionals, was lacking in other studies. For example, 150 children, 20 parents and 10 teachers were included in the validation study by Cardon et al. [31]. These participants were asked to provide qualitative feedback on the clarity and relevance of the items. People who are required to review and critique a data collection tool need to be experts in the topic of interest [45]. The inclusion of experts in judging the quality of a research instrument is vital as these are the best individuals to assess whether the items are able to measure what they are supposed to measure. For example, they can suggest adding relevant items or removing unnecessary variables. 

Of all the studies found in the literature [11,12,13,14,15,16,17,18,19,20,21,22,23,24,25,26,27,28,29,30,31], only two studies, by Sirajudeen and Siddik [19] and Noll et al. [30], reported the quantitative results of the validation study. Content validity of the questionnaire used by Sirajudeen and Siddik [19] was earlier established by Sirajudeen et al. [46]. Similar to the current study, a mixed methods (quantitative and qualitative) approach was used to validate the research instruments [30,46]. Nine experts (orthopaedic surgeons, physiotherapists, research methodology expert, psychiatrist, community health physician and information technology expert) provided their feedback regarding accuracy, relevance and appropriateness of the content in Sirajudeen et al.’s study [46]. The overall content validity index was found to be 0.98, indicating high validity. The judges also suggested minor amendments regarding the clarity or wording of the items, and those revisions were incorporated into the instrument. This finding was in agreement with the current study where the overall content validity index of the adult version of the Aristegui questionnaire was found to be 0.96 (very close to 0.98).

Eight experts in back pain, body posture or biomechanics of human movement were asked to judge the content validity of the Back Pain and Body Posture Evaluation Instrument (BackPEI) [30]. The experts scored each item as being “well suited”, “suitable” or “unsuitable” regarding the clarity and relevance of the items. Although the sample size was larger (eight compared to five in the present study) [30], they used the percentage of “well suited” responses to analyse the collected data. The authors found this percentage to be 98.2%. The participants also provided their qualitative feedback on the clarity and wording of the tool items.

A major drawback of using the percentage agreement as an estimate of the content validity is that it does not consider the possibility of chance agreement. There is always a possibility that some of the judges scored at least some items by guessing, in particular when they were not sure which score to provide [47]. Thus, the observed agreement could be considered to be a false agreement leading, therefore, to flawed results. Consequently, the current study used the content validity index of each item and of the whole scale as well as the Kendall coefficient of concordance, thereby resulting in increased reliability of the results.

The test–retest reliability demonstrated very low variability between the two tests, suggesting therefore that the adult version of the Aristegui questionnaire could be considered reliable in assessing the knowledge regarding ergonomics and posture as well as postural behaviour in university students. Almost perfect agreements were found for the items evaluating knowledge of adequate body posture when sitting at a desk (Qs 12) (k = 1.00, *p* = 0.00), on an armchair (Qs 19) (k = 0.80, *p* = 0.01), at a table (Qs 21) (k = 1.00, *p* = 0.01) and using computer (Qs 23) (k = 1.00, *p* = 0.01). This suggests a high level of agreement between the subjects’ responses in two periods with a one-week interval. These items seemed to be easy to remember by participants in the retest that took place after 7 days. This can be interpreted by the fact that university students perform these activities in daily living, suggesting that these variables could be appropriate for the evaluation of knowledge of university students regarding ergonomics and posture as well as their postural behaviour.

Substantial agreement was shown for the questions on knowledge on picking up objects from the ground (Qs 10) (k = 0.63, *p* = 0.01) and carrying backpacks (Qs 15) (k = 0.78, *p* = 0.00). This could be attributed to the fact that the test might have increased the students’ interest to learn more about the body posture. The question on knowledge regarding sleeping posture (Qs 8) had a moderate agreement (k = 0.51, *p* = 0.03). The question on knowledge on carrying a bag on wheels (Qs 25) had a fair agreement (k = 0.29, *p* = 0.19). This was probably because it is hard for an individual to adopt one standard posture during sleeping, thus it becomes hard to have consistency in the responses on this item after a one-week period. Moreover, bags on wheels are often used for travel and not frequently used at university so it could be considered hard to remember.

Most of the instruments found in the literature [11,12,13,14,15,16,17,18,19,20,21,22,23,24,25,26,27,28,29,30,31] that measured knowledge regarding ergonomics and posture as well as postural behaviour were more complicated with longer questionnaires in comparison to the adult version of the Aristegui questionnaire. In addition, Noll et al.’s study [30] was applied to a schoolchild population whose level of understanding was lower than that of the university students studied here, thereby limiting the comparison of the results. In contrast to the previous Likert scale questionnaires, the adult version of the Aristegui questionnaire is a pictorial multiple choice questionnaire and this led to variability in the statistical methods used in the data analysis. For example, Sirajudeen and Siddik [19] used a questionnaire that had been earlier tested for reliability by Sirajudeen and Pillai [7]. Using a similar design, 20 computer professionals were asked to complete the questionnaire twice with a period of 2 weeks in between [7]. 

To evaluate the consistency of participants’ responses over time, the authors used Pearson’s correlation coefficient (r) according to the following criteria: high reliability for r > 0.90, good reliability for r between 0.80 and 0.89, fair reliability between 0.70 and 0.79 and poor reliability for r < 0.70 [7]. The results revealed that correlation values were 0.79 for working at the computer, 0.84 for sitting, 0.77 for keyboard/mouse, 0.75 for monitor and 0.76 for table and accessories. Furthermore, the correlation values were highly significant for all the items (*p* < 0.001). Thus, the questionnaire demonstrated fair to good reliability. Although the statistical analysis and criteria differed between Sirajudeen and Pillai’s study [7] and the current study, the results from both studies revealed acceptable validity for the items in common. These were sitting at a desk (Qs 12) (k = 1.00, *p* = 0.00), on an armchair (Qs 19) (k = 0.80, *p* = 0.01), at a table (Qs 21) (k = 1.00, *p* = 0.01) and using computer (Qs 23) (k = 1.00, *p* = 0.01). 

On the other hand, the study by Noll et al. [30] assessed the consistency of the Back Pain and Body Posture Evaluation Instrument (BackPEI). The 21-item questionnaire was developed to determine the prevalence of back pain and its risk factors (sociodemographic, genetic, lifestyle and postural behaviour). To test the stability of the instrument, 260 school children were administered the test and retest with a one-week period in between. In agreement with the findings of the present study, Noll et al. reported the following kappa values [30]: for the posture adopted when sitting at a desk (k = 0.64, *p* = 0.001), sitting on a chair (k = 0.65, *p* = 0.002), sitting down working at the computer (k = 0.64, *p* = 0.001), picking up objects from the ground (k = 0.65, *p* = 0.001) as well as carrying a bag (k = 1.00, *p* = 0.001). 

Noll et al. [30] also used similar classifications for the kappa coefficient. Reliability was categorised as poor (k < 0.2), fair (0.2 < k < 0.4), moderate (0.4 < k < 0.6), good (0.6 < k < 0.8) or very good (k > 0.8). Although they observed higher kappa values for postural behaviour when picking up objects from the floor (0.65 compared to 0.00–0.62) [30], the current study also found acceptable values of kappa coefficient for this posture. The higher reliability estimates of lifting posture observed by Noll et al. [30] in comparison with the present study could be related to the larger sample size or the age of the target population.

This study is not without limitations. The findings of the validation part were based on the evaluation of five out of ten experts. Although a sample of ten would have been more representative of the intended professionals, only five agreed to participate in the study. Future research may need to ask more than 10 judges and probably use other ways of communication in addition to the emails (telephone) in order to increase both the response rate as well as the size of the sample. The use of a convenience sample together with the small number of respondents in the test–retest reliability might have led to a less representative sample for university students. The small sample size might have not ensured that the reproducibility of the results would be achieved when repeating the study. More research with a larger sample would be useful to further verify the consistency of the adult version of the Aristegui questionnaire. Therefore, it is recommended that the results are interpreted with caution and the current study is replicated in future by using a larger and more representative sample of experts.

## 5. Conclusions

The originality of this study lies in the fact that this is the first study to evaluate the extent to which the adult version of the Aristegui questionnaire can be considered to be a valid and reliable tool in assessing knowledge regarding ergonomics and posture as well as postural behaviour in university students. Five experts with good knowledge about the topic scored the relevance of each item and provided qualitative feedback on the comprehensiveness and clarity of the items as well as suggestions regarding minor or major amendments. 

The results of this study revealed that the overall content validity index of the adult version of the Aristegui questionnaire was 0.96 revealing, therefore, a high level of content validity. Furthermore, the results demonstrated slight to perfect reliability of the questionnaire for most of the dichotomous and multiple choice items involved. It can be concluded that the adult version of the Aristegui questionnaire is a valid and reliable tool to measure university students’ knowledge regarding ergonomics and posture as well as postural behaviour. The use of this valid and reliable instrument in future studies may help explore the association between back care knowledge and the presence of musculoskeletal pain as well as the effect of a back care educational intervention on the university student population’s knowledge and behaviour. Further studies with larger and more diverse samples should be considered to confirm the findings of this study.

## Figures and Tables

**Figure 1 healthcare-12-00324-f001:**
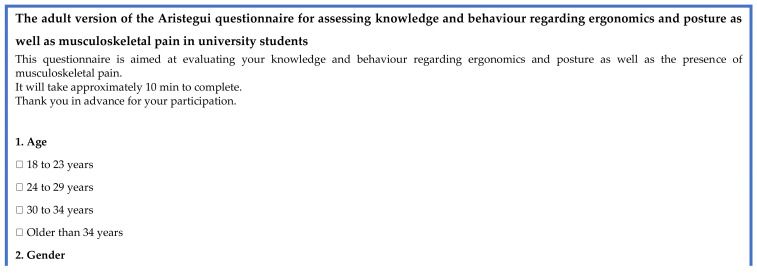
The adult version of the Aristegui questionnaire.

**Figure 2 healthcare-12-00324-f002:**
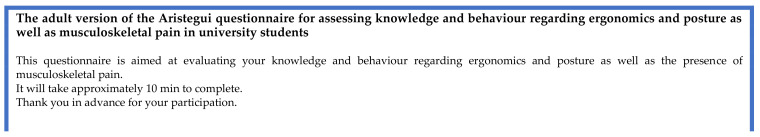
The content validated questionnaire.

**Figure 3 healthcare-12-00324-f003:**
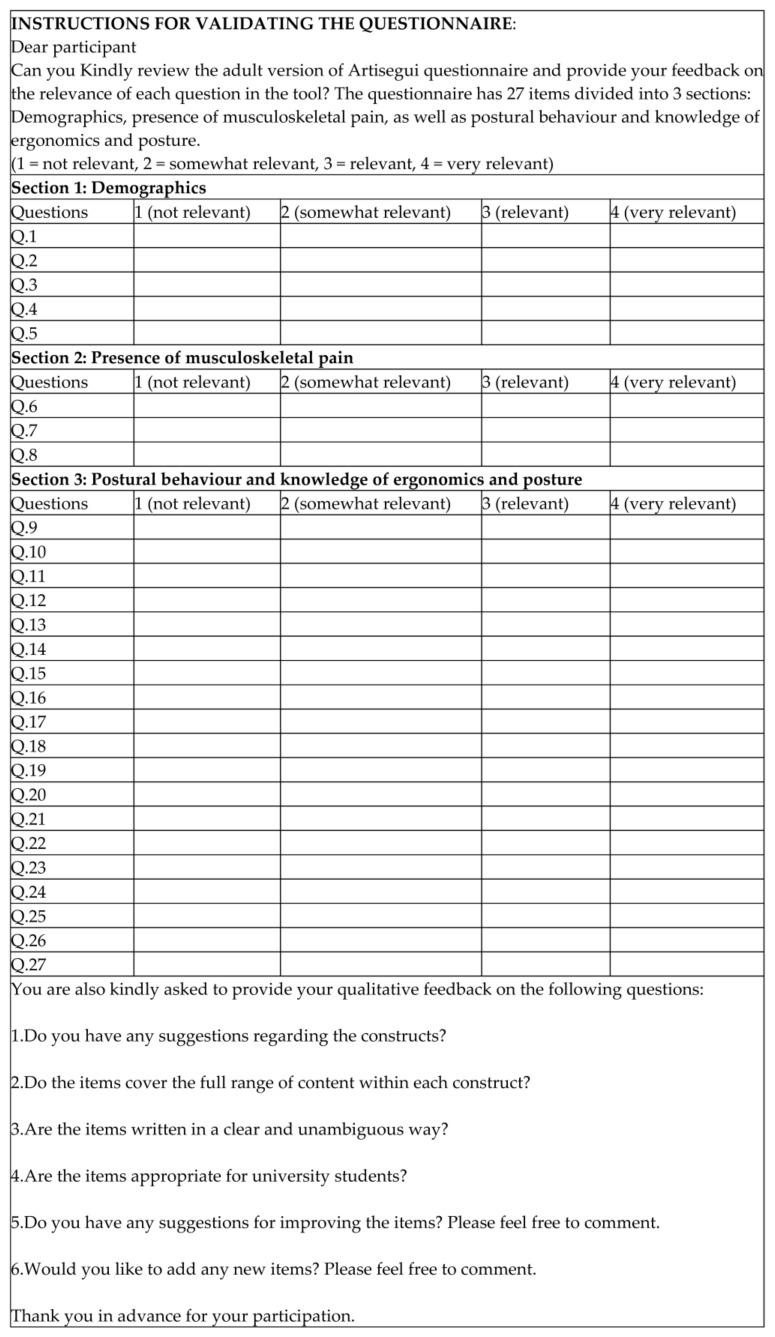
Content validation form.

**Table 1 healthcare-12-00324-t001:** Results of the content validity study.

Items	Expert A	Expert B	Expert C	Expert D	Expert E	Number of Agreement	I-CVI	Decision
1. Age	4	4	4	4	4	5	1	Appropriate
2. Gender	3	4	4	4	3	5	1	Appropriate
3. Course of study	3	4	3	3	3	5	1	Appropriate
4. Course level	2	4	3	2	1	2	0.4	Eliminated
5. Academic year	2	4	3	2	1	2	0.4	Eliminated
6. Have you ever had musculoskeletal pain?	4	4	3	4	4	5	1	Appropriate
7. Where have you had musculoskeletal pain?	4	4	4	4	4	5	1	Appropriate
8. After what activity do you experience musculoskeletal pain?	4	4	1	4	4	4	0.8	Appropriate
9. Which of the following positions do you adopt when lying down?	4	4	3	4	4	5	1	Appropriate
10. What do you think is the most adequate sleeping posture?	3	4	3	4	4	5	1	Appropriate
11. Which of the following positions do you adopt when picking up objects from floor?	4	4	3	4	4	5	1	Appropriate
12. What do you think is the most adequate posture when picking up objects from floor?	4	4	3	4	4	5	1	Appropriate
13. Which of the following positions do you adopt when sitting at a desk?	4	4	3	4	4	5	1	Appropriate
14. What do you think is the most adequate posture when sitting at desk?	4	4	3	4	4	5	1	Appropriate
15. Do you carry backpack when you attend university?	4	4	3	4	4	5	1	Appropriate
16. Which of the following positions do you adopt when carrying a backpack?	4	4	3	4	4	5	1	Appropriate
17. What do you think is the most adequate posture when carrying a backpack?	4	4	3	4	4	5	1	Appropriate
18. Which of the following positions do you adopt when taking objects from high shelf?	2	4	3	4	4	4	0.8	Appropriate
19. What do you think is the most adequate posture when taking objects from high shelf?	2	4	3	4	4	4	0.8	Appropriate
20. Which of the following positions do you adopt when sitting on armchair?	4	4	3	3	4	5	1	Appropriate
21. What do you think is the most adequate posture when sitting on armchair?	4	4	3	3	4	5	1	Appropriate
22. Which of the following positions do you adopt when sitting at table?	4	4	3	3	4	5	1	Appropriate
23. What do you think is the most adequate posture when sitting at table?	4	4	3	3	4	5	1	Appropriate
24. Which of the following positions do you adopt when sitting down working at computer?	4	4	3	3	4	5	1	Appropriate
25. What do you think is the most adequate posture when sitting down working at computer?	4	4	3	3	4	5	1	Appropriate
26. Which of the following positions do you adopt when carrying a bag on wheels?	3	4	3	4	4	5	1	Appropriate
27. What do you think is the most adequate posture when carrying a bag on wheels?	2	3	3	3	4	4	0.8	Appropriate

**Table 2 healthcare-12-00324-t002:** Characteristics of the sample in the test–retest study.

Respondents’ Characteristics	Frequency and Percentage N (%)
Age	
18 to 23 years	1 (10%)
24 to 29 years	3 (30%)
30 to 34 years	1 (10%)
Older than 34 years	5 (50%)
Gender	
Male	6 (60%)
Female	4 (40%)
Course of study	
Biomedical sciences	1 (10%)
Business	1 (10%)
Computer science	2 (20%)
Education	1 (10%)
Electric engineering	1 (10%)
Law	1 (10%)
Microbiology	1 (10%)
Psychology	1 (10%)
Public health	1 (10%)

**Table 3 healthcare-12-00324-t003:** Test–retest reliability results for the adult version of the Aristegui questionnaire.

Item Number and Description	Percentage Agreement (%)	Measure of Agreement Kappa (k)	Standard Error (SEk)	Significance Level (*p*-Value)	95% CI
1. Age	1.00	1.00	0.00	0.00	1.00–1.00
2. Gender	1.00	1.00	0.00	0.00	1.00–1.00
3. Course of study	1.00	1.00	0.00	0.00	1.00–1.00
4. Have you ever had musculoskeletal pain?	1.00	1.00	0.00	0.00	1.00–1.00
(5a) Neck and shoulder pain	1.00	0.60	0.23	0.04	0.54–1.05
(5b) Upper back pain	1.00	1.00	0.00	0.00	1.00–1.00
(5c) Low back pain	0.80	0.62	0.23	0.04	0.17–1.05
(6a) Musculoskeletal pain after lying down	0.90	0.74	0.24	0.02	0.27–1.20
(6b) Musculoskeletal pain after picking up objects from floor	0.90	0.62	0.34	0.04	0.04–1.27
(6c) Musculoskeletal pain after sitting	0.80	0.55	0.26	0.05	0.04–1.04
(6d) Musculoskeletal pain after standing	0.90	0.62	0.34	0.04	0.04–1.27
(6e) Musculoskeletal pain after carrying weight	1.00	1.00	0.00	0.00	1.00–1.00
(6f) Musculoskeletal pain after using computer	0.80	0.80	0.19	0.01	0.44–1.16
(6g) Musculoskeletal pain after walking around	1.00	1.00	0.00	0.00	1.00–1.00
(7a) Sleep on the back	1.00	1.00	0.00	0.00	1.00–1.00
(7b) Sleep on the stomach	0.90	0.74	0.24	0.02	0.27–1.20
(7c) Sleep on the side	0.70	0.44	0.22	0.09	0.01–0.88
(7d) Sleep on back, stomach and side	0.80	0.55	0.26	0.05	0.04–1.05
8. What do you think is the most adequate sleeping posture	0.70	0.51	0.23	0.03	0.05–0.96
(9a) Pick up objects by bending with flexion of knees and spine	0.80	0.60	0.23	0.04	0.54–1.05
(9b) Pick up objects with both back and knees straight	0.70	0.40	0.28	0.20	0.16–0.96
(9c) Pick up objects by bending with flexion of knees	0.80	0.00	0.00	1.00	0.00–0.00
(9d) Pick up objects by bending with flexion of knees and spine, back and knees straight and flexion of knees	0.90	0.62	0.34	0.04	0.04–1.27
10. What do you think is the most adequate posture when picking up objects from ground	0.90	0.63	0.32	0.01	0.00–1.26
(11a) Sit at a desk with spine on the back of chair	0.80	0.41	0.30	0.11	0.18–1.00
(11b) Sit at a desk with pelvic retroversion	0.90	0.74	0.24	0.02	0.27–1.20
(11c) Sit at a desk with spine leaning forwards	0.80	0.52	0.29	0.10	0.05–1.10
(11d) Sit at a desk with all of the above	0.90	0.78	0.20	0.01	0.39–1.18
12. What do you think is the most adequate posture when sitting at a desk	1.00	1.00	0.00	0.00	1.00–1.00
13. Do you carry a backpack when attending university	1.00	1.00	0.00	0.00	1.00–1.00
(14a) Carry the backpack on both shoulders close to the body	1.00	1.00	0.00	0.00	1.00–1.00
(14b) Carry the backpack on both shoulders far from the body	0.90	0.74	0.24	0.02	0.27–1.20
(14c) Carry the back on one shoulder	0.90	0.74	0.24	0.02	0.27–1.20
(14d) Carry the backpack on both shoulders, close to body, far from body, and one shoulder	0.90	0.62	0.34	0.04	0.04–1.27
15. What do you think is the most adequate posture when carrying a backpack	0.90	0.78	0.17	0.00	0.44–1.12
(16a) Take objects from a shelf with hyperextension of the back	0.80	0.60	0.25	0.06	0.10–1.10
(16b) Take objects from a shelf with use of a step	1.00	1.00	0.00	0.00	1.00–1.00
(16c) Take objects from a shelf with extension of arms	0.80	0.41	0.30	0.11	0.18–1.00
(16d) Take objects from a shelf with hyperextension of back, use of step and extension of arms	0.90	0.62	0.34	0.04	0.05–1.28
17. What do you think is the most adequate posture when taking objects from shelf	0.90	0.63	0.32	0.01	0.00–1.25
(18a) Correct alignment when sitting on armchair	0.90	0.00	0.00	1.00	0.00–0.00
(18b) Leaning forwards when sitting on armchair	0.90	0.00	0.00	1.00	0.00–0.00
(18c) Pelvic retroversion when sitting on armchair	0.80	0.60	0.25	0.06	0.10–1.10
(18d) Correct alignment, leaning forwards and pelvic retroversion when sitting on armchair	0.90	0.80	0.19	0.01	0.44–1.16
19. What do you think is the most adequate posture when sitting on armchair	0.60	0.80	0.19	0.01	0.44–1.16
(20a) Correct alignment when sitting at a table	0.8	0.38	0.36	0.24	0.33–1.08
(20b) Leaning forwards when sitting at a table	0.9	0.35	0.30	0.26	0.24–0.93
(20c) Pelvic retroversion when sitting at a table	0.8	0.38	0.36	0.24	0.33–1.08
(20d) Correct alignment, leaning forwards and pelvic retroversion when sitting at a table	0.8	0.58	0.26	0.06	0.07–1.10
21. What do you think is the most adequate posture when sitting at a table	1.00	1.00	0.00	0.00	1.00–1.00
(22a) Leaning forwards when using computer	0.9	0.74	0.24	0.02	0.27–1.20
(22b) Correct alignment when using computer	0.8	0.52	0.29	0.09	0.05–1.10
(22c) Pelvic retroversion when using computer	1.00	1.00	0.00	0.00	1.00–1.00
(22d) Leaning forwards, correct alignment and pelvic retroversion when using computer	1.00	1.00	0.00	0.00	1.00–1.00
23. What do you think is the most adequate posture when using computer	1.00	1.00	0.00	0.00	1.00–1.00
(24a) Carry bag on wheels with hand facing backwards and arm close to body	0.8	0.00	0.00	1.00	0.00–0.00
(24b) Carry bag on wheels with hand facing forwards and arm close to body	0.8	0.54	0.26	0.05	0.04–1.05
(24c) Carry bag on wheels with hand facing backwards and arm far from the body	0.8	0.11	0.08	0.73	0.04–0.26
(24d) Carry bag on wheels with hand facing backwards and arm close to body, hand facing forwards and arm close to body, hand facing backwards and arm far from the body	0.8	0.00	0.00	1.00	0.00–0.00
25. What do you think is the most adequate posture when carrying a bag on wheels	0.6	0.29	0.19	0.19	0.10–0.67

## Data Availability

The datasets generated during and/or analyzed during the current study are available from the corresponding author upon reasonable request.

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
