# Peer review of "Development of a Novel Pictorial Questionnaire to Assess Knowledge and Behaviour on Ergonomics and Posture as Well as Musculoskeletal Pain in University Students: Validity and Reliability"

_healthcare, 2024, doi:10.3390/healthcare12030324_

Round 1
Reviewer 1 Report
Comments and Suggestions for Authors
Hello,
Your paper is very clearly written and easy to follow. No significant concerns.
Best of luck!
* Page 14- rather than listing out the formulas, are you able to simply reference each statistic/equation, or just provide results?
* Out of curiosity, what is the age range for your > 34 y/o population?
Reviewer 2 Report
Comments and Suggestions for Authors
Dear authors,
The subject is interesting and corresponds perfectly to the themes of the journal. The originality of the proposed questionnaire should be highlighted and detailed in relation to questionnaires in the international literature. I feel it's important to take into account minor questions and comments in order to improve the article :
- In the abstract, the authors state that the questionnaire was evaluated by 5 experts (line 18), whereas in the sampling section (line 113) 10 experts appear to have been used (5 experts in physiotherapy, 2 experts in sports therapy, 1 in nursing, 1 chiropractor and 1 podiatrist). Which information is correct?
- In the introduction, the small review proposed on line 45 should be detailed, as it is very important in demonstrating the interest and relevance of the proposed questionnaire. The works listed should be presented, highlighting the objectives and results of each, so that the reader can assess the relevance of the proposed questionnaire in relation to existing ones. The authors did not mention the work of jacquier et al 2023 on a postural taxonomy (Smart-taxo addressing posture, sitting, lying and walking) applied to the evaluation of MSDs during smartphone use among university students, those of Odole et al. (2020), who investigated patterns of musculoskeletal pain, postural abnormalities, and smartphone usage among undergraduate university students and that of Gorce et al 2023 (cross-sectional survey) on postural prevalence, time of day and spent time activities during smartphone weekday use among student. The presentation of these studies would demonstrate the contribution and relevance of the questionnaire proposed in the article.
- J Jacquier-Bret, P Gorce, Effect of day time on smartphone use posture and related musculoskeletal disorders risk: a survey among university students, BMC Musculoskeletal Disorders 24 (1), 2023
- Adesola C Odole1, Dorcas A Olutoge1*, Oluwagbohunmi Awosoga2, Chidozie E Mbada3, Clara Fatoye4, Olufemi O Oyewole5, Ruth I Oladele1, Francis A Fatoye4 and Aderonke, O Akinpelu1 Musculoskeletal Pain and Postural Abnormalities among, Smartphone-Using Digital Natives, J Musculoskelet Disord Treat, 2020, 6:089,
- P Gorce, J Jacquier-Bret, Postural prevalence, time of day and spent time activities during smartphone weekday use among students: A survey to prevent musculoskeletal disorders, Heliyon 9 (12) , 2023
- In the sampling section - it seems important to me to add the years of experience of each of the experts in their discipline for greater clarity and so that the reader has all the information. Why aren't there any biomechanical or ergonomic experts, despite their in-depth training in posture characterization and its impact on MSDs?
- Line 485 - Why didn't you justify the number of students taking part in the validation procedure?
- In the discussion section, the authors could focus on the originality of the questionnaire and discuss this point.
Sincerely,
